# Efficacy of Boron Neutron Capture Therapy in Primary Central Nervous System Lymphoma: In Vitro and In Vivo Evaluation

**DOI:** 10.3390/cells10123398

**Published:** 2021-12-02

**Authors:** Kohei Yoshimura, Shinji Kawabata, Hideki Kashiwagi, Yusuke Fukuo, Koji Takeuchi, Gen Futamura, Ryo Hiramatsu, Takushi Takata, Hiroki Tanaka, Tsubasa Watanabe, Minoru Suzuki, Naonori Hu, Shin-Ichi Miyatake, Masahiko Wanibuchi

**Affiliations:** 1Department of Neurosurgery, Osaka Medical and Pharmaceutical University, 2-7 Daigaku-Machi, Takatsuki 569-8686, Japan; kohei.yoshimura@ompu.ac.jp (K.Y.); hideki.kashiwagi@ompu.ac.jp (H.K.); yusuke.fukuo@ompu.ac.jp (Y.F.); koji.takeuchi@ompu.ac.jp (K.T.); gen.futamura@ompu.ac.jp (G.F.); ryo.hiramatsu@ompu.ac.jp (R.H.); wanibuchi@ompu.ac.jp (M.W.); 2Institute for Integrated Radiation and Nuclear Science, Kyoto University, 2 Asashiro-Nishi, Kumatori-cho, Sennan 590-0494, Japan; taku-takata@rri.kyoto-u.ac.jp (T.T.); tanaka.hiroki.3e@kyoto-u.ac.jp (H.T.); watanabe.tsubasa.8x@kyoto-u.ac.jp (T.W.); suzuki.minoru.3x@kyoto-u.ac.jp (M.S.); 3Kansai BNCT Medical Center, Osaka Medical and Pharmaceutical University, 2-7 Daigaku-Machi, Takatsuki 569-8686, Japan; naonori.ko@ompu.ac.jp (N.H.); shinichi.miyatake@ompu.ac.jp (S.-I.M.)

**Keywords:** boron neutron capture therapy (BNCT), malignant brain tumor, primary central nervous system lymphoma (PCNSL), radiation therapy

## Abstract

Background: Boron neutron capture therapy (BNCT) is a nuclear reaction-based tumor cell-selective particle irradiation method. High-dose methotrexate and whole-brain radiation therapy (WBRT) are the recommended treatments for primary central nervous system lymphoma (PCNSL). This tumor responds well to initial treatment but relapses even after successful treatment, and the prognosis is poor as there is no safe and effective treatment for relapse. In this study, we aimed to conduct basic research to explore the possibility of using BNCT as a treatment for PCNSL. Methods: The boron concentration in human lymphoma cells was measured. Subsequently, neutron irradiation experiments on lymphoma cells were conducted. A mouse central nervous system (CNS) lymphoma model was created to evaluate the biodistribution of boron after the administration of borono-phenylalanine as a capture agent. In the neutron irradiation study of a mouse PCNSL model, the therapeutic effect of BNCT on PCNSL was evaluated in terms of survival. Results: The boron uptake capability of human lymphoma cells was sufficiently high both in vitro and in vivo. In the neutron irradiation study, the BNCT group showed a higher cell killing effect and prolonged survival compared with the control group. Conclusions: A new therapeutic approach for PCNSL is urgently required, and BNCT may be a promising treatment for PCNSL. The results of this study, including those of neutron irradiation, suggest success in the conduct of future clinical trials to explore the possibility of BNCT as a new treatment option for PCNSL.

## 1. Introduction

Boron neutron capture therapy (BNCT) is a nuclear reaction-based tumor cell-selective particle irradiation that selectively destroys tumor cells and has been clinically applied as a treatment for invasive cancers, such as high-grade meningiomas [1] and gliomas [2,3,4]. In BNCT, a boron compound is delivered directly to the tumor cells prior to the therapy, which accumulates in the tumor cells. Subsequent neutron irradiation results in the capture of boron-10 (^10^B) atoms by thermal neutrons to produce high linear energy transfer particles (alpha particles and ^7^Li recoil nuclei). Because these particles have a short path length (5–9 μm), and their path length roughly corresponds to the size of a single tumor cell (10 μm), they only destroy B-containing cells. Owing to its limited effect within a single cell, it can selectively destroy tumor cells and preserve normal cells. According to the World Health Organization, primary central nervous system lymphoma (PCNSL) is a grade IV malignant brain tumor that accounts for approximately 5% of all brain tumors in Japan. PCNSL is a common disease in older adults, and 62% of patients with this tumor are aged >60 years [2,3]. The number of patients with PCNSL has increased in recent years [4]. The prognostic factors for PCNSL are age and general performance status [5], and the 2-year survival rate ranges from 50% to 80% to 15% in the malignant group [6]. The treatment strategies for PCNSL are selected based on the results of the biopsy, which are used to confirm the pathological diagnosis of the tumor, and include chemotherapy with high-dose methotrexate (HD-MTX) and whole-brain radiation therapy (WBRT). PCNSL has a very high response rate to initial therapy, but relapse and recurrence occur in almost all patients. Because of the overlap of adverse events associated with the initial therapy, only a few effective treatments are available for recurrent PCNSL; therefore, the prognosis of recurrent PCNSL is poorer. In a previous study using lymphoma cells, lymphoma cells showed a higher sensitivity to photon radiation than other cancer cell types and that the cytotoxicity of BNCT was higher than that of photon irradiation [7]. Therefore, the high cell selectivity and therapeutic effect of BNCT could be expected in patients with refractory and recurrent PCNSL. This is because this cell-selective irradiation treatment can be combined safely with radiotherapy or delivered as a re-irradiation method [8]. To explore the therapeutic application of BNCT for PCNSL, we aimed to conduct a basic study on BNCT using a human lymphoma cell line and a mouse central nervous system (CNS) lymphoma brain tumor model and to evaluate the efficacy and safety of BNCT for PCNSL. This is the first study to evaluate the efficacy of in vivo BNCT for PCSNL.

## 2. Materials and Methods

### 2.1. Cell Culture

Raji and RL human lymphoma cell lines have been histologically characterized as human Burkitt’s lymphoma. The Raji lymphoma cells and RL cells were purchased from the Japanese Collection of Research Bioresources and American Type Culture Collection, respectively. Roswell Park Memorial Institute (RPMI) 1640 medium containing 10% fetal bovine serum (FBS) and 1% penicillin-streptomycin were used for cell cultures. The cells were grown at 37 °C in a humidified incubator with 5% CO_2_ until they reached confluence. Cell viability was determined by trypan blue dye staining. Cultures with more than 90% cell viability were used in this study. All materials for cell culture were purchased from Gibco Invitrogen (Grand Island, NY, USA).

### 2.2. Borono-Phenylalanine

4-Borono-L-phenylalanine (BPA) (L-isomer), which was used as a boron carrier in this study, was kindly supplied by Stella Pharma (Osaka, Japan) and converted to a fructose complex [9]. The BPA used in this study was ^10^B enriched (96% enriched in ^10^B atoms).

### 2.3. Boron Uptake in Human Lymphoma Cells

In this study, we examined whether exposure to BPA resulted in boron uptake into the human lymphoma cells, Raji and RL. First, 1 × 10^5^ Raji and RL cells/mL were seeded in 100-mm dishes (Becton, Dickinson, and Company, Franklin Lakes, NJ, USA) at 37 °C in a 5% CO_2_ atmosphere using the culture medium described above. After 1 day of incubation at 37 °C, the medium was replaced with a culture medium containing 5, 10, and 20 μg B/mL of BPA, followed by incubation at 37 °C for 3 h. The medium containing BPA was removed, and the dishes were washed twice with phosphate-buffered saline (PBS). Then, PBS was added and centrifuged twice to count and sediment the cells. The cells were then lysed overnight in a 1N nitric acid solution. The boron concentration in the cells was measured by inductively coupled plasma atomic emission spectroscopy (ICP-AES; Hitachi, Tokyo, Japan).

### 2.4. Photon Irradiation Study of Human Lymphoma Cells

In this study, we calculated the cell viability ratio of two types of human lymphoma cells, Raji and RL, after photon irradiation to investigate the effect of photon radiation. Specifically, the lymphoma cells, Raji and RL, were cultured for 3 days each and then irradiated at dose rates of 2, 4, 8, 12, 16, and 32 Gy using a photon radiation device (M-150WE; SOFTEX, Tokyo, Japan). After photon irradiation, the human lymphoma cells were seeded in 96-well plates at 1.0 × 10^5^ cells/well. After 96 h of incubation at 37 °C, the cell viability ratio was calculated using the water-soluble tetrazolium-8 (WST-8) assay with a Cell Counting Kit-8 (CCK-8, Dojindo, Kumamoto, Japan). The cell viability ratio was calculated using a modified version of the manufacturer-provided protocols [10] and according to Equation (1):Cell viability ratio (%) = (absorbance value of irradiated cell/absorbance value of untreated cells) × 100(1)

### 2.5. Neutron Irradiation Study of Human Lymphoma Cells

In this study, the lymphoma cells, Raji and RL, were cultured for several days and then exposed to BPA for 3 h, followed by neutron irradiation for 0, 15, and 30 min in a research reactor in KURNS. Neutron irradiation was performed at a reactor power of 1 MW and neutron flux of 6.0 × 10^8^ neutrons/cm^2^/s for 15 and 30 min. The physical dose values of thermal, epithermal, fast neutrons, and γ rays, respectively, in this irradiation of 15 and 30 min were 0.11, 0.02, 0.11, and 0.2 Gy for a total of 0.5 Gy and 0.24, 0.03, 0.18, and 0.72 Gy for a total of 1.2 Gy. The concentrations of BPA administered were 5, 10, and 20 μg/mL. After neutron irradiation, the cells were collected and plated at 1.0 × 10^5^ cells/well in 96-well plates. After 96 h of incubation at 37 °C, the cell viability ratio was measured using the WST-8 assay with CCK-8. The cell viability ratio was calculated by modifying the protocol provided by the manufacturer and further corrected based on the results of the photon irradiation study.

### 2.6. Mouse CNS Lymphoma Model

All animal studies were performed in accordance with the Guide for the Care and Use of Laboratory Animals and approved by the Animal Use Review Board and Ethical Committee of Osaka Medical College (permit no. 2020-107) and the Institute for Integrated Radiation and Nuclear Science, Kyoto University (KURNS; Kumatori, Osaka, Japan) (permit no. 2020-31). The animals used for the study were eight-week-old male BALB/c nu/nu mice (Japan SLC; Shizuoka, Japan). Each mouse was anesthetized by an intraperitoneal injection of a mixture of three different anesthetic agents: medetomidine (ZENOAQ, Fukushima, Japan) (0.4 mg/kg), midazolam (Sandoz, Yamagata, Japan) (2.0 mg/kg), and butorphanol (Meiji Seika, Tokyo, Japan) (5.0 mg/kg). The animal’s head was fixed with a stereotactic frame (IMPACT-1000C + MA-625, Muromachi Kikai, Tokyo, Japan). The Raji lymphoma cells were surgically implanted into the mouse brain using a technique that was previously adopted by our research group to confirm the efficacy of a novel boron compound. The Raji lymphoma cells were diluted in a 10 μL solution of RPMI containing 1.4% agarose (Wako Pure Chemical Industries, Osaka, Japan) at a concentration of 2.0 × 10^5^ cells and were injected at a rate of 20 μL/min using an automatic infuser pump. The mice were implanted with Raji lymphoma cells in the brain and sacrificed when the endpoint was reached. The microscopic examination of the mouse brain revealed that the tumors had formed in the intracerebral region. The histopathological evaluation showed lymphoma cells in the brain parenchyma, with the diffuse proliferation of small blasts similar to human PCNSL (Figure A1). In this CNS lymphoma model using the Raji human lymphoma cell line, tumor engraftment was achieved, and the model was determined to be suitable for further therapeutic studies to evaluate the effect of BNCT.

### 2.7. In Vivo Biodistribution Study

After tumor implantation, when the mice showed signs of tumor growth (i.e., weight loss, lethargy, hunching, and ataxia), in vivo biodistribution studies were initiated. BPA was administered to the mouse CNS lymphoma model at doses equivalent to those of boron and was eventually adjusted to 24 mg B/kg body weight (b.w.). Three hours later, the mice were euthanized, and the tumor, brain, blood, heart, lung, liver, spleen, kidney, and skin were removed and weighed. The amount of boron in each organ was quantified by ICP-AES. 

### 2.8. In Vivo Irradiation Study

Neutron irradiation studies were performed to investigate the therapeutic effect of neutron irradiation after BPA administration in the Raji mouse CNS lymphoma model. The mice transplanted with 2.0 × 10^5^ Raji lymphoma cells were treated with neutron irradiation at KURNS. Neutron irradiation was performed at a reactor power of 1 MW and neutron flux of 8.1 × 10^8^ neutrons/cm^2^/s for 15 and 30 min. Thirty-five mouse CNS lymphoma models were randomly divided into five groups. Irradiation was performed 3 h after the intraperitoneal administration of BPA (24 mg B/kg b.w.).
Group 1: Untreated controlGroup 2: BPA onlyGroup 3: Neutron irradiation onlyGroup 4: 15min of neutron irradiation following BPA administration (BNCT 15-min group)Group 5: 30min of neutron irradiation following BPA administration (BNCT 30-min group)

After the animals were anesthetized using a mixture of the abovementioned anesthetic agents, their bodies (except for the head) were covered with ^6^LiF ceramic tiles to protect the thermal neutrons in order to reduce whole-body exposure to radiation. Neutron irradiation was performed 3 h after BPA administration. After the neutron irradiation study, the mouse CNS lymphoma model was left at KURNS. The survival time of the mice in all the groups was used to evaluate the treatment effect. Based on the MST, we calculated and evaluated the percentage of increased lifespan (%ILS) for each group relative to the untreated control, as reported in our previous studies [11].

### 2.9. Statistical Analysis

Statistical analyses were performed using the JMP Pro 15 software (SAS Institute Inc., Cary, NC, USA). Student’s *t*-test was performed to compare the cell viability ratio. A log-rank test was performed to test the equality of survival curves of in vivo irradiation study. Finally, a *p*-value of less than 0.05 was considered significant.

## 3. Results

### 3.1. Boron Uptake in Human Lymphoma Cells

The cellular boron concentrations obtained using BPA, incubated with 5, 10, and 20 µg B/mL, are shown in the graph (Figure 1). In 5 μg B/mL BPA, both Raji and RL showed high concentrations (64.4 ± 4.7 and 71.4 ± 2.5 μg B/10^9^ cells). Even in 10 μg B/mL of BPA, both Raji and RL showed high concentrations (107 ± 14.4 and 108.4 ± 7.6 μg B/10^9^ cells). Meanwhile, 20 μg B/mL of BPA showed similar results (187.5 ± 7.9 and 205.7 ± 8.4 μg B/10^9^ cells).

### 3.2. Photon Irradiation Study of Human Lymphoma Cells

The cell viability ratio at 96 h after photon irradiation is shown in Figure 2. In Raji cells, the cell viability ratio decreased when the irradiation dose was increased from 2 Gy to 32 Gy (0.62 ± 0.07, 0.58 ± 0.07, 0.28 ± 0.06, 0.19 ± 0.03, 0.19 ± 0.04, and 0.13 ± 0.03). A statistically significant decrease in the cell viability ratio was observed at 2 Gy (*p* = 0.0008), and the same observation was reported when the irradiation dose was increased to 32 Gy. The *p*-values from 4 to 32 Gy were as follows: *p* = 0.0004, 0.00002, 0.00004, 0.00001, and 0.00002. In RL cells, the cell viability ratio decreased when the irradiation dose was increased from 2 to 32 Gy (0.72 ± 0.03, 0.33 ± 0.02, 0.23 ± 0.01, 0.17 ± 0.007, 0.14 ± 0.01, and 0.18 ± 0.01) and showed a significant difference (*p* = 0.02). The *p*-values from 4 to 32 Gy were as follows: *p* = 0.0007, 0.0005, 0.0004, 0.0002, and 0.0001.

### 3.3. Neutron Irradiation Study of Human Lymphoma Cells

The cell viability ratio at 96 h after neutron irradiation is shown in Figure 3. BNCT after 5 μg B/mL exposure significantly reduced the cell viability ratio in Raji lymphoma cells compared with that in controls (0.23 ± 0.07 vs. 1.00 ± 0.13, *p* = 0.0001), and the same result was observed in RL (0.33 ± 0.01 vs. 1.00 ± 0.02 vs. *p* = 0.02). The results were similar for both Raji and RL at other concentrations: 10 μg B/mL (0.12 ± 0.03; *p* = 0.0004 and 0.29 ± 0.02; *p* = 0.04) and 20 μg B/mL (0.14 ± 0.03; *p* = 0.0003 and 0.25 ± 0.04; *p* = 0.01).

### 3.4. In Vivo Biodistribution Study

The boron concentrations in the tumor, brain, blood, and other organs were measured. The boron concentrations in the tumor, brain, and blood were 9.9 ± 1.6, 3.5 ± 1.8 µg/g, and 13.8 ± 3.5 µg/mL, respectively, and the boron concentration in the tumor was significantly higher than that in the brain (*p* = 0.0006). The tumor to normal brain (contralateral brain) ratio was 2.84, while the tumor to blood ratio was 0.72. Other organs, such as the heart, lungs, liver, kidney, spleen, and skin, were also measured, and results showed the accumulation of boron (13.0 ± 2.8, 19.4 ± 6.4, 14.4 ± 6.2, 53.2 ± 17.4, 23.6 ± 8.0, and 18.7 ± 9.4 µg/g). The data are summarized in Figure 4.

### 3.5. In Vivo Irradiation Study

The Kaplan–Meier survival curves for the mouse CNS lymphoma model that underwent neutron irradiation are shown in Figure 5. The MST durations for the control group (untreated control, BPA only, neutron irradiation only) and the BNCT group (BNCT 15-min or BNCT 30-min) were 26.5 (range: 21–39 days), 31.5 (28–39 days), 31 (22–33 days), 44 (33–36 days), and 39 days (36–40 days). The survival time was significantly longer in the BNCT 30-min group (*p* = 0.007) than in the untreated control group and between the BNCT 15-min group and the untreated control group (p = 0.02). In addition, the BNCT 30-min group showed the highest %ILS value among all groups (66.0%). The %ILS values for the other groups (BPA only, neutron only, and BNCT 15-min) were 17.0%, 18.9%, and 47.2%, respectively (Table 1).

## 4. Discussion

The results of in vitro boron uptake studies showed that both human lymphoma cell lines, Raji and RL, showed a concentration-dependent increase in cellular boron uptake upon exposure to BPA (Figure 1). BPA is taken up into the cells via the L-type amino acid transporter 1 (LAT1), an amino acid transporter specifically expressed in cancer cells [12]. In addition, previous studies have shown that tumor cells from PCNSL express high levels of LAT1 [13]. Based on these facts and the results of this study, BPA is a suitable boron-carrying drug used in BNCT for the treatment of PCNSL and that a high concentration of BPA in the tumor would result in a higher therapeutic effect.

Photon irradiation studies showed that lymphoma cells, Raji and RL, showed dose-dependent cytotoxicity and high radiosensitivity (Figure 2). The neutron irradiation study showed a significant decrease in the cell viability ratio after neutron irradiation and BPA exposure (Figure 3). This result proved that in lymphoma cells, the cell viability ratio was significantly reduced by BNCT in vitro. However, despite the increase in cellular boron concentration in the in vitro boron uptake study (Figure 1), a concentration-dependent decrease in cell viability ratio was not observed in the neutron irradiation study (Figure 3). Based on these results, we considered that the concentration of boron administered and the concentration of boron actually taken up into the cells were different. Actually, it has been reported that the uptake of BPA differs depending on the cell cycle [14]. We estimated the biological photon equivalent dose (equivalent dose) in the in vitro neutron irradiation studies and calculated the combined biological effectiveness (CBE) value of BPA in each lymphoma cell. Specifically, a logistic regression analysis of the photon irradiation study results was performed. The equivalent dose in the neutron irradiation study was estimated by applying the cell survival rate of neutron irradiation to the results of the logistic analysis of photon irradiation. The calculated CBE value of the boron compounds was calculated by applying the estimated equivalent dose to the following equation, which has been proposed in many previous studies. D_B_ is the physical dose of boron derived from Equation (2). The relative biological effectiveness (RBE) for nitrogen (RBE_N_) value was 3.0, and that for hydrogen (RBE_H_) value was also 3.0 [15].
Photon equivalent dose (Gy-Eq) = D_B_ × CBE + D_N_ × RBE_N_ + D_H_ ×RBE_H_ + D_γ_(2)

7.43 × 10^−14^ (Gy cm^2^/μg B/g) × boron concentration (μg B/g) × thermal neutron fluence (1/cm^2^). DN is the physical dose of nitrogen derived from the equation 6.78 × 10^−14^ (Gy cm^2^/weight %) ×  nitrogen concentration (weight %) × thermal neutron fluence (1/cm^2^). Dγ is the physical dose of gamma radiation. These values were obtained from a previous study [16]. In this study, the DN value was 0.43, the DH value was 0.24, and the Dγ value was 0.72. The calculated CBE value to a boron concentration of 5 μg B/mL in Raji cells was 10.6, with an estimated equivalent dose value (photon-equivalent dose; Gy-Eq) of 11.2. The equivalent doses of 10 and 20 μg B/mL in Raji cells could not be estimated due to deviations from the results of the logistic analysis. The calculated CBE value to a boron concentration of 5 μg B/mL in RL cells was 5.6 with an equivalent dose value of 7.2 (Gy-Eq), the calculated CBE value to a boron concentration of 10 μg B/mL was 3.4 with an estimated equivalent dose value of 5.4 (Gy-Eq), and the calculated CBE value to a boron concentration of 20 μg B/mL was 2.0 with an estimated equivalent dose value of 6.4 (Gy-Eq). In RL cells, the equivalent dose did not increase in a boron concentration-dependent manner, and the estimated CBE values decreased. There are two reasons for this result (a) if the cells were washed with PBS and placed in the medium without boron for irradiation, BPA leaked from the cells, and there was no or little boron in the cells during neutron irradiation; therefore, the results were pretty much the same for all boron concentration groups; (b) the neutron beam was contaminated with gamma rays and fast neutrons to the extent where boron neutron capture reaction result was masked or hidden in the more prominent effect of the beam components. This finding suggests that the CBE values estimated in this study may not be accurate. The CBE value of gliomas currently using BNCT in clinical practice is 3.8 [17]; by contrast, the CBE value of lymphomas estimated in this study was higher than that value. The two types of lymphoma cells used in this study were floating cells, so we could not perform the colony formation assay typically used to study the effects of photon irradiation. Therefore, we compared the results of the photon irradiation and neutron irradiation studies by correcting for cell viability as measured by the WST-8 assay. As a supplement, fitting graphs of the linear-quadratic (LQ) model using the cell viability obtained by the WST-8 assay are shown in the Appendix (Figure A2). BNCT for lymphoma caused a decrease in the cell viability ratio. Therefore, we believe that this study demonstrates the major advantage of BNCT for PCNSL in vitro.

In this study, we performed in vitro cell studies using Raji and RL cells. We created a mouse CNS lymphoma model using Raji cells, which has been previously reported in mouse CNS lymphoma models. Mouse CNS lymphoma models have been used in various studies [18], and other lymphoma models have been established in previous studies. Wang et al. created a CNS lymphoma model using Raji cells similar to the one we used in this study and showed that celecoxib prolonged their survival [19]. Since the RL cells showed good boron uptake in vitro and BNCT showed a significant decrease in cell viability, in vivo irradiation studies were not performed. We created a mouse CNS lymphoma model using RL cells, but the tumor implantation rate was low. Hence, the model was not used in the study as we thought it was unsuitable for neutron irradiation studies using a small number of animal models in the limited environment of a nuclear reactor.

The biodistribution in the mouse CNS lymphoma model showed that a higher BPA concentration was taken up in the tumor than in the normal brain (Figure 4). Other organs also showed BPA uptake, and the highest uptake values were found in the kidneys. Previous studies have shown that BPA is rapidly cleared by the kidneys in mouse models and in clinical human trials [20]. In this study, the kidneys retained high concentrations of BPA, suggesting that the drug was also cleared by the urinary system in the mouse lymphoma model. Neutron irradiation studies using a mouse CNS lymphoma model showed a significant prolongation of survival in the BNCT group compared with the control group, and no significant difference was observed in the neutron irradiation only group (Figure 5). The reason why survival time was prolonged only in the BNCT group and not in the neutron irradiation group was thought to be due to the selective uptake of boron into the tumor, which allowed more equivalent doses to be delivered to the tumor in the BNCT group than in the neutron irradiation only group. This may be due to the fact that tumor-selective boron uptake was obtained in the mouse CNS lymphoma model, as revealed by biodistribution studies. Therefore, we determined the equivalent radiation dose for the mouse CNS model using the CBE value calculated based on the data from the in vitro study (Table 2).

As shown in the table, the equivalent doses delivered to the tumor were 8.8 and 16.6 (Gy-Eq) compared with 1.4 (Gy-Eq), while those delivered to the normal brain were 1.0 and 2.0 (Gy-Eq) compared with 1.4 (Gy-Eq) in the neutron irradiation only and BNCT groups, respectively. This result indicates that the tumor was irradiated with a higher equivalent dose compared to the brain, even though the same neutrons were irradiated, indicating that the tumor-selective treatment of BNCT was indeed achieved.

PCNSL was initially treated with WBRT because of its high sensitivity to photon radiation, but the use of MTX in the 1970s improved the patients’ survival [21]. Later, PCNSL was found to be highly sensitive to HD-MTX, and the mainstay of treatment was a combination of HD-MTX and WBRT. The treatment strategies for PCNSL are selected based on the results of the biopsy, which is performed to confirm the pathological diagnosis of the tumor and include chemotherapy using high-dose methotrexate (HD-MTX) and WBRT [22]. The response rate to initial therapy is nearly 90%, and the median survival is approximately 36 months, indicating a good response to initial therapy [23]. However, the recurrence rate is high, and the prognosis of patients with PCNSL who have recurrence is poor, with a median survival of 2 months without additional therapy. Most PCNSL patients are aged >60 years; the incidence of leukoencephalopathy is significantly increased when a combination of MTX therapy and WBRT is used in patients aged >60 years compared with HD-MTX therapy alone [24,25]. Several studies have also observed a significant decrease in Karnofsky Performance Status and a significant decrease in quality of life (QOL) in older patients after undergoing WBRT [26,27]. The mechanism of WBRT-induced neurotoxicity is not yet clear, but it is thought to involve vascular toxicity, demyelination, and depletion of neural progenitor cells from the subventricular zone [28]. PCNSL has a high recurrence rate relative to the initial treatment response rate; therefore, recurrence is inevitable, but effective treatment options for PCNSL that recur after treatment have not yet been established. Additional radiotherapy is not recommended due to the possibility of delayed neurotoxicity. There is no effective treatment for recurrent PCNSL because the use of a combination of two treatments increases the possibility of adverse events; hence, a new treatment must be developed. If boron is selectively distributed to the tumor, BNCT can provide an extremely large equivalent dose to the tumor but a sufficient dose to the normal tissue, which is irradiated with a neutron beam [29]. In other words, it is possible to provide a dose to normal tissues that do not cause adverse events and a sufficient dose to tumors. This property of BNCT is very beneficial for recurrent PCNSL, for which treatment is limited due to the occurrence of adverse events caused by photon irradiation. Therefore, when BNCT is used in the clinical setting, these benefits will be assessed from the treated patients, and we will conduct a clinical trial targeting the treatment of recurrent PCNSL. In addition, since BNCT has high tumor selectivity and excellent local control [30], it can improve the QOL by resolving the clinical symptoms of recurrent PCNSL, such as paralysis. In addition, by performing BNCT as an initial treatment, it is possible to perform HD-MTX or WBRT without worrying about the possible occurrence of leukoencephalopathy at the time of recurrence. We believe that brain edema and radiation-induced late effects due to neutron irradiation after photon irradiation may raise a concern when BNCT is performed as a treatment for recurrent PCNSL. With regard to this issue, previous clinical studies on BNCT have not reported any new adverse events due to BNCT when used as a treatment for recurrent glioblastoma after radiotherapy [8]. Of course, it is difficult to compare the different diseases that require treatment; however, we believe that this is one of the reasons why the safety of using BNCT for recurrent PCNSL after radiotherapy. In the future, we will conduct repeat irradiation studies on PCNSL to confirm its efficacy and safety.

Due to the use of a nuclear reactor in this study, neutron irradiation experiments could only be performed 3 h after BPA administration. Therefore, it is possible that different results could be obtained if the experiments were conducted at different times. In addition, the photon irradiation in this study was performed at a machine with the maximum photon energy of 150 keV, which is smaller than the 250 keV of a standard. Therefore, if a different irradiation device is used, the results may differ.

## 5. Conclusions

We performed basic research on BNCT using a mouse CNS lymphoma brain tumor model and a human lymphoma cell line. This study suggests that BNCT could potentially be a promising treatment for PCNSL and showed a high possibility of contributing to the future development of PCNSL treatment and BNCT.

## Figures and Tables

**Figure 1 cells-10-03398-f001:**
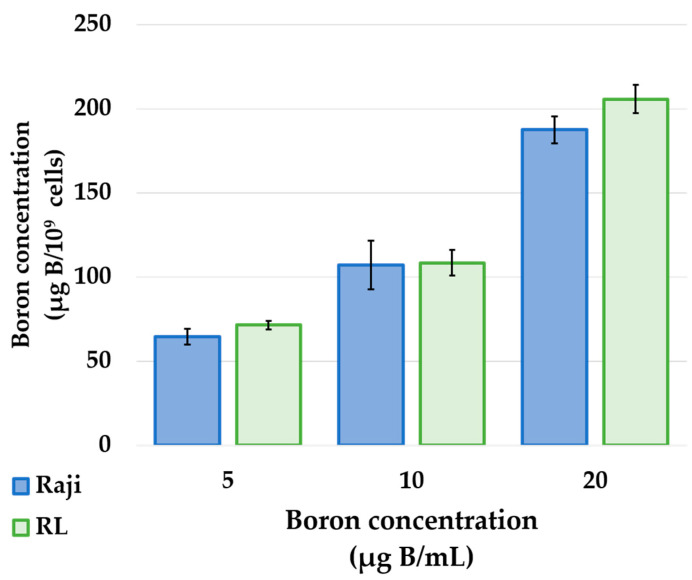
Cellular uptake of boron in human lymphoma cells. Boron concentration of Raji (blue) and RL (green) lymphoma in RPMI 1640, using 5, 10, and 20 µg B/mL of BPA for 3 h. The boron concentration of lymphoma cells was high for both Raji and RL.

**Figure 2 cells-10-03398-f002:**
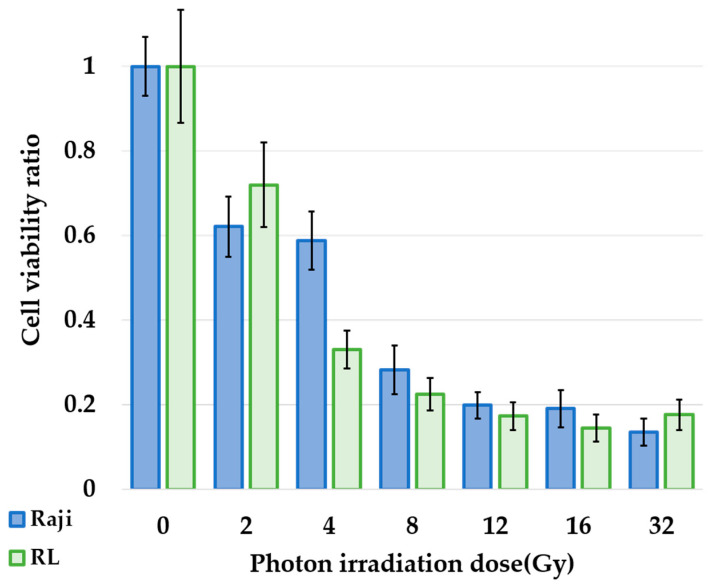
Relationship between cell viability ratio and photon radiation dose. The cell viability ratio of lymphoma cells after 96 h of irradiation from 0 to 32 Gy. Cell viability ratio of Raji cells are shown in blue, and that of RL are shown in green. There was a significantly decreased cell viability ratio by more than 2 Gy irradiation compared to 0 Gy in Raji and RL (*p* < 0.05).

**Figure 3 cells-10-03398-f003:**
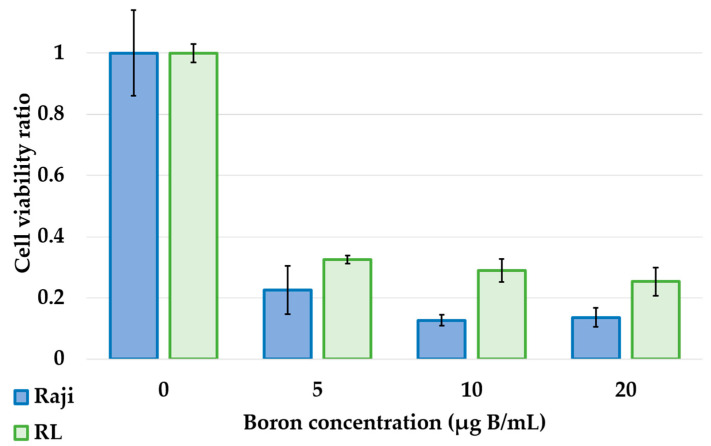
Relationship between boron concentration and cell viability ratio after neutron irradiation. Cell viability of lymphoma cells after exposure to BPA at concentrations of 5, 10, and 20 μg B/mL for 3 h followed by irradiation with 1 MW neutrons for 30 min and incubation for 96 h. Blue is Raji lymphoma cells, and green is RL lymphoma cells. In Raji lymphoma cells. BNCT significantly reduced the cell viability of lymphoma cells (*p* < 0.05).

**Figure 4 cells-10-03398-f004:**
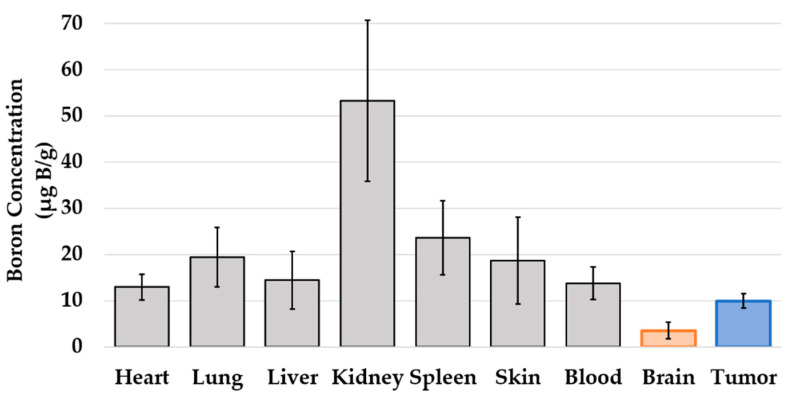
Boron concentrations in various organs of mouse CNS lymphoma models treated with BPA. Tumors had significantly higher boron concentrations (9.9 ± 1.6 µg/g) compared to the brain (3.5 ± 1.8 µg/g: *p* < 0.05). The tumor to brain ratio (: T/Br) was 2.85. The tumor to blood ratio (: T/Bl) was 0.72. Boron accumulation was also observed in other organs.

**Figure 5 cells-10-03398-f005:**
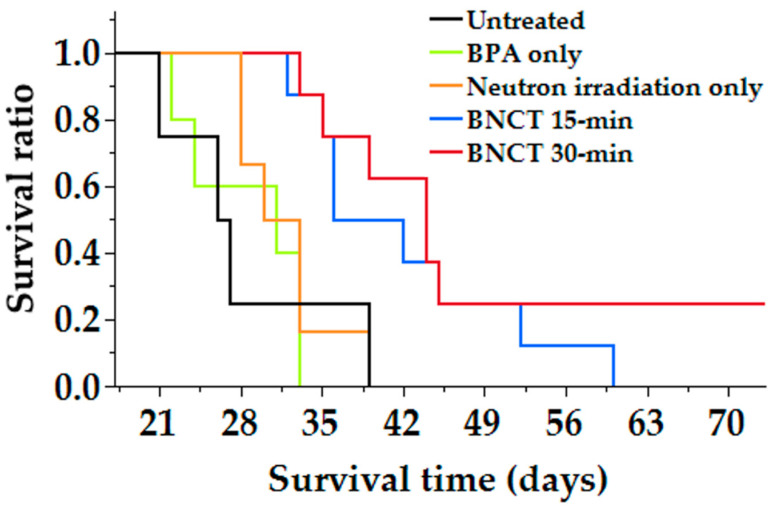
Kaplan–Meier survival curves of mouse CNS lymphoma models after neutron irradiation. Survival times in days after tumor implantation were plotted for the following groups: untreated control (black line), BPA only (green line), neutron irradiation only (orange line), BNCT 15-min: BPA exposure followed by 15 min of neutron irradiation (blue line), and BNCT 30-min: BPA exposure followed by 30 min of neutron irradiation (brown line). Median survival times of untreated control group, BPA only group, Neutron irradiation only group, BNCT 15-min group, and BNCT 30-min group were 28.2 ± 3.8, 28.6 ± 2.3, 31.8 ± 1.7, 42.3 ± 3.4, and 41.3 ± 1.8 days. There were statistically significant differences between the BNCT 30-min group and untreated control group (*p* = 0.007) and between the BNCT 15-min group and untreated control group (*p* = 0.02).

**Table 1 cells-10-03398-t001:** Survival times of mouse CNS lymphoma models after neutron irradiation.

Group	Irradiation Time (min)	*n* ^1^	Survival Time			%ILS ^2^
			Mean ± SD	Median	Range	Median
Untreated control	0	6	28.2 ± 3.8	26.5	23.5–33	
BPA only	0	6	28.6 ± 2.3	31	24–33	17.0%
Neutron irradiation only	30	8	31.8 ± 1.7	31.5	28–33	18.9%
BNCT 15-min	15	7	42.3 ± 3.4	39	35.5–48.5	47.2%
BNCT 30-min	30	8	41.3 ± 1.8	44	37–	66.0%

^1^ Number of animals. ^2^ Percent increased life span (%ILS) was defined relative to the mean survival times of untreated controls.

**Table 2 cells-10-03398-t002:** Summary of physical radiation doses and equivalent doses delivered in Raji lymphoma bearing mice.

Group	Physical Dose ^1^		Equivalent Dose ^2^	
	Brain	Tumor	Brain	Tumor
Untreated control	0	0	0	0
BPA only	0	0	0	0
Neutron irradiated only	0.6	0.6	1.4	1.4
BNCT 15-min	0.3	0.3	1.0	8.8
BNCT 30-min	0.6	0.6	2.0	16.6

^1^ Physical dose estimates include doses from gamma rays, ^10^B(*n*, alpha)^7^Li, ^14^N(*n*, *p*)^14^C, and ^1^H(*n*, *n*)^1^H reactions. ^2^ The equivalent dose is a value calculated by the equation. D_B_ × CBE_B_ + D_N_ × RBE_N_ + D_H_ × RBE_H_ + D_γ_. (D_B_: Boron dose, D_N_: neutron dose, D_H_: hydrogen dose, D_γ_: gamma-ray dose). CBE_B_ is based on the Compound Biological Effectiveness (CBE) of D_B_, and in the case of BPA, this value is 10.6 for tumor tissue and 0.9 for the normal brain. RBE_N_ is based on RBE (Relative Biological Effectiveness) for D_N_, and this value is 3.0. RBE_H_ is based on RBE (Relative Biological Effectiveness) for D_H_, and this value is 3.0.

## Data Availability

The data that support the findings of this study are available from the corresponding author, S.K., upon reasonable request.

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
