# Peer review of "Efficacy of Boron Neutron Capture Therapy in Primary Central Nervous System Lymphoma: In Vitro and In Vivo Evaluation"

_cells, 2021, doi:10.3390/cells10123398_

Round 1

Reviewer 1 Report

The manuscript of Kohei Yoshimura et al. successfully proposes an application of Boron Neutron Capture Therapy for Primary Central Nervous System Lymphoma.
The work is clearly written, the results have been presented with adequate methodologies and the conclusions are consistent with the results obtained, therefore it deserves publication on Cells after correcting linguistic errors and subjecting the work to a certified mother tongue proofreading.

Author Response

We are grateful for the time and energy you expended on our behalf. We wish to express our appreciation to the Reviewer for the insightful comments again. These comments have helped to significantly improve this manuscript. We trust that the revised manuscript is suitable for publication.

Reviewer 2 Report

Reviewer’s comments

In the manuscript by Yoshimura et al., the authors describe in vitro and in vivo boron neutron capture therapy (BNCT) experiments on human primary central nervous system lymphoma (PCNSL) and discuss the feasibility of this method for further clinical application based on the obtained results. The idea of this study is clear, and the irradiation experiments are done in a standard manner which speaks to the importance of this study and its place in the overall development of BNCT as a method. However, several issues need to be considered before further evaluation of this manuscript.

  1. The title doesn’t comply with the study described in the manuscript. The title is more suited to an extensive clinical study, although the authors have conducted several experiments that are still far from clinical. Please add “in vitro” and “in vivo” in the title. For example, it can sound as follows: “Efficacy of Boron Neutron Capture Therapy in Primary Central Nervous System Lymphoma in Vitro and in Vivo,” or “In Vitro and in Vivo Efficacy of Boron Neutron Capture Therapy in Primary Central Nervous System Lymphoma,” or similar to these.
  2. English grammar seems to be fine, but the style should be corrected appropriately, as many sentences with words with the same meaning, definition repetitions are present in the manuscript. For example, on line 23: “The cellular boron concentration in human lymphoma cells…”; subtitles in the sections, line 85: “In Vitro…..in …..Cells”; line 156: “In Vivo Irradiate(?) Study of Mouse…”; line 159: “The mice…were treated in vivo…”, etc. As a different issue fo style or meaning, on line 121 and further in the text: “Finally, a p-value of less than 0.05 was considered significant” might be interpreted as before the final time point, other p-values were considered significant, which is confusing. And this sentence is repeated several times at the end of different paragraphs when it could be mentioned once in the Statistical Analysis paragraph.
  3. The categorical statement that “BNCT is a promising treatment for PCNSL” should be avoided, as the authors use the results of their single in vivo irradiation experiment to support their statement. The words, such as “might” or “may,” will sound less categorical and more appropriate in this case.
  4. In the Introduction, the authors should state the novelty of this study clearly; otherwise, using a different tumor model and different tumor cell lines in reactor-based irradiation experiments might look similar to dozens of other in vitro and in vivo preclinical evaluations of BNCT effects. The authors might use the following phrases to state the novelty: “We are not aware of other preclinical studies on human PCNSL treatment by BNCT……”, or “To the best of our knowledge, we are the first to….”.
  5. In the Materials and Methods in subsection 2.2 Boron Compounds (actually there was only one boron compound, so the name of the subsection should be changed), it would be better to indicate the percentage of boron-10 in the overall boron amount in BPA, in other words, the purity of the isotope, which might be critical for the absorbed dose evaluation.
  6. The authors incubated cells for 3 hours in BPA solution and injected BPA 3 hours before mice irradiation. Could they justify this timing? BPA accumulation is cell type- and time-dependent, and at least several time points should be assessed to verify the highest boron accumulation. If this study is unique and no prior experiments were done on human lymphoma cells, how did the authors know that 3 hours is the best timing to achieve the most appropriate boron concentration? I agree that repeating the experiments with our time points might be difficult in this case; however, the authors should indicate the reason for their choice. This and further unclear issues should be described in the limitations of the study in the Discussion section.
  7. Regarding the cell experiments, the authors describe cell washing with PBS and adding PBS for cell counting. If the cells were kept in PBS for some time before further centrifugation and addition of the nitric acid for cell lysis, some part of BPA could escape into PBS, and the concentration could be different from reality. Similarly, if the cells were kept in the medium without boron before and during irradiation, the BPA could easily leak from the cells, and the BNCT effect could have been significantly reduced by the low concentration or absence of boron, as shown in the neutron irradiation experiments in this manuscript.
  8. The description of cell irradiation experiments lacks some important information, including the abovementioned issue. Could the authors add the data on the neutron flux and the overall fluence? Could the authors provide the data of the neutron beam components, including thermal, epithermal, fast neutrons, gamma contamination? The results of the neutron irradiation could simply be related to high gamma or fast neutron doses, as well as the washed-out boron, as mentioned above.
  9. Photon irradiation was done at the M-150WE device, which has a 150kV tube, and the mean energy of the photons should be about 70 keV with the major impact of about 20 to 70 keV photons. All this is slightly different from 250keV X-rays used as a comparative standard in irradiation studies. There might be almost no difference, of course, and the effect can be the same. However, the authors might have discussed this issue as well as X-ray filtration, where using aluminum or copper filer will shift the spectrum of photons and will influence radiation absorption by cells and, therefore, cell survival. The authors may refer to the X-ray spectrum of the tube they used. Probably, this comment is too strict for this paper, but the authors should at least discuss these practical issues, as they may change the results of the experiments considerably and influence the flow of other dependent experiments. The following references may help to study this issue more precisely:

Poludniowski GG. Calculation of x-ray spectra emerging from an x-ray tube. Part II. X-ray production and filtration in x-ray targets. Med Phys. 2007;34(6):2175-2186. http://dx.doi.org/10.1118/1.2734726

Zaboronok A, Tsurushima H, Yamamoto T, Isobe T, Takada K, Sakae T, Yoshida F, Matsumura A. Size-dependent radiosensitization effects of gold nanoparticles on human U251 malignant glioma cells. Nanosci. Nanotechnol. Lett. 5, 990-994. 2013. http://dx.doi.org/10.1166/nnl.2013.1646

(https://www.researchgate.net/publication/275434792_Size-Dependent_Radiosensitization_Effects_of_Gold_Nanoparticles_on_Human_U251_Malignant_Glioma_Cells)

You can study the issue of X-ray spectrum used in your study based on software simulation with a free educational software SpekCalc: http://spekcalc.weebly.com/

to model what X-rays you were using in your photon irradiation experiments.

  1. As an issue for further theoretical background in these experiments, the cell cycle may play a role in BPA accumulation and the results of neutron irradiation experiments. Please, consider this as well.
  2. Regarding the photon irradiation, in the sentence on lines 117-119 “The cell viability ratio was calculated by modifying the protocol provided by the manufacturer and further corrected based on the results of the photon irradiation study.” it is not clear how the results of neutron irradiation experiments were corrected using the results of photon experiment and why.
  3. Boron accumulation histograms for two cell lines can be easily united in one graph for better visualization (Figure 1). The corresponding data (boron concentrations) are given twice in the Results section and in the Figure 1 legend. Please leave one set, as the repetition is not necessary. In other figure legends, the repetition of the results is not necessary.
  4. Could the authors explain why they used a 96-hour interval to assess the cell survival after irradiation and the colorimetric method? Typically, a colony-forming assay 10-14 days after the irradiation is used to show longer-term effects of radiation, which are more informative. 3-day evaluation might be too early to fully evaluate cell survival. Please give justification.

Another issue is the visualization method (Figure 2). Typically, cell survival curves are plotted as linear graphs and are fit to the linear-quadratic equation models based on the radiobiological parameters (alpha and beta) calculated from the fit. The exponential decrease in the cell survival after neutron irradiation can be compared to that after photon or other irradiation, using the absorbed doses, for example; however, the direct comparison might be impossible, and in the case of neutron irradiation, using neutron fluence or boron concentration might be more representative.

For LQ-model fit and radiobiological parameters calculation, the authors could use regression methods in SPSS or a much simpler method provided by a SOLVER add-on in Microsoft Excel. Please follow this link for a more detailed explanation of the curve fit in Excel: https://www.youtube.com/watch?v=Ewp5CF5ba_w

To statistically verify the difference between the cell survival curves, between the two cell lines, for example, the area under the curve (AUC) for each exponential curve should be calculated for each of the three independent irradiation experiments, and the mean values with standard deviations might be used to find if the difference is statistically significant (by ANOVA, Student t-test or another method).

These references might help to plot the cell survival curves and calculate the radiobiological parameters alpha and beta, and other dependent parameters, either for photon or neutron irradiation:

Franken, N. A., Rodermond, H. M., Stap, J., Haveman, J., & van Bree, C. (2006). Clonogenic assay of cells in vitro. Nature protocols, 1(5), 2315–2319. https://doi.org/10.1038/nprot.2006.339

Sato, E., Zaboronok, A., Yamamoto, T., Nakai, K., Taskaev, S., Volkova, O., Mechetina, L., Taranin, A., Kanygin, V., Isobe, T., Mathis, B. J., & Matsumura, A. (2018). Radiobiological response of U251MG, CHO-K1 and V79 cell lines to accelerator-based boron neutron capture therapy. Journal of radiation research, 59(2), 101–107. https://doi.org/10.1093/jrr/rrx071

Zaboronok, A.; Taskaev, S.; Volkova, O.; Mechetina, L.; Kasatova, A.; Sycheva, T.; Nakai, K.; Kasatov, D.; Makarov, A.; Kolesnikov, I.; Shchudlo, I.; Bykov, T.; Sokolova, E.; Koshkarev, A.; Kanygin, V.; Kichigin, A.; Mathis, B.J.; Ishikawa, E.; Matsumura, A. Gold Nanoparticles Permit In Situ Absorbed Dose Evaluation in Boron Neutron Capture Therapy for Malignant Tumors. Pharmaceutics 2021, 13, 1490. https://doi.org/10.3390/pharmaceutics13091490

  1. In Figure 4 legend, it might be better to add Tumor/Blood ratio, as it might be essential in neutron irradiation experiments for tissue exposure estimation.
  2. Regarding animal experiments and the Kaplan-Meier curves in Figure 5, please check the colors of the legend, as they might not correspond to the data in Table 1 and in the figure legend. According to the colors, the green (BPA only) group lived longer than other even treated groups.
  3. In the Discussion, the authors tried to justify the results of in vitro neutron irradiation experiments, which seemed to be not so informative compared to photon irradiation experiments. However, all the calculations of the photon equivalent doses here might not be relevant due to several reasons: a) if the cells were washed with PBS and placed in the medium without boron for irradiation, BPA leaked from the cells, and there was no or little boron in the cells during neutron irradiation; therefore, the results were pretty the same for all boron concentration groups; b) the neutron beam was contaminated with gamma rays and fast neutrons to the extent where boron neutron capture reaction result was masked or hidden in the more prominent effect of the beam components. I understand that irradiation experiments at the reactor can’t be easily repeated, but for future experiments, the authors need to consider all the conditions and variations of experiments in order to obtain informative results. Please avoid unnecessary discussions and add limitations of the study to the Discussion.

“The major advantage of BNCT for PCNSL in vitro” hasn’t been shown as proper cell survival curves have not been built, and radiobiological parameters have not been calculated. Please refer to the corresponding literature to build the survival curves and provide proof of the efficacy.

  1. Please provide references at the end of the sentences on lines 369 (“…patients’ survival [?].) and 373 (“…and WBRT [?].).
  2. Please verify the sentence “In this study, the kidneys retained high concentrations of BPA, suggesting that the drug was also cleared by the urinary system in the rat lymphoma model.” (lines 339-341), as it says about the “rat lymphoma model” but not mice used in this study. Or should there be “as” put in the sentence? “as in the rat lymphoma model,” for example? In such a case, the proper reference to the source should be given.
  3. This study is a pilot study, and significant work has to be done to transfer BNCT for PCNSL from this stage to the clinical level, therefore, giving the statements that this study is a “preliminary stage of the clinical development and application of BNCT as treatment for PCNSL…” might be too preliminary. It would be better to use less categorical statements and put the conclusions on what the authors actually found in their study. Also, the decision if the findings are “extremely important” or not will be made by the readers. Please rewrite the Conclusion accordingly.

Otherwise, it is a valuable study, which is important for further development of BNCT and its spread in the oncology field, and after the recommendations are taken into account, and the manuscript is properly revised, it can be suitable for publication.

Round 2

Reviewer 2 Report

The authors have adequately answered the reviewer’s questions. However, there are still some issues that should be better corrected before publication.

The sentence in the Figure 2 legend is confusing, as the cell viability is rate/a number, but not the color: “cell viability ratio of Raji cells was blue and that of RL was green.” Please rewrite. In Figures 1 and 3, for example, it is clear, and no corrections are needed.

Could the authors add small legends on the graphs in Figures 1,2, and 3 for better visualization, as they have done in Figure A2? For example, blue square for Raji cells and green square for RL.

Could the authors exclude “Cellular” from the titles of the paragraphs to avoid repetitions?

“2.3. Cellular Uptake of Boron in Human Lymphoma Cells” to

“2.3. Boron Uptake in Human Lymphoma Cells”? and

“3.1. Cellular Uptake of Boron in Human Lymphoma Cells” to

“3.1. Boron Uptake in Human Lymphoma Cells”?

If the authors decided to build the X-ray machine spectrum, they should correct it based on the M-150WE irradiator parameters, where the maximum tube voltage is 150 but not 80kVp.

https://www.softex-kk.co.jp/products/m_we.html

And rewrite the sentence:

“In addition, the mean energy of the photon in this study is about 80 keV, which is smaller than the 250 keV of a standard.”

into

“In addition, the photon irradiation in this study was done at the machine with the maximum photon energy of 150 keV, which is smaller than the 250 keV of a standard”.
